# Membrane potential modulates ERK activity and cell proliferation in human cells

Mari Sasaki*, Masanobu Nakahara, Takuya Hashiguchi, Fumihito Ono

Department of Physiology, Division of Life Science, Faculty of Medicine, Osaka Medical and Pharmaceutical University, Takatsuki, Japan

## eLife Assessment

This **important** article employs multiple experimental approaches and presents evidence that changes in membrane voltage directly affect ERK signaling to regulate cell division. This result is relevant because it supports an ion channel-independent pathway by which changes in membrane voltage can affect cell growth. The evidence now presented is **solid** and the data support the conclusions. This article should be of interest to a broad readership in the areas of cell and developmental biology and electrophysiology.

*For correspondence:
mari.sasaki@ompu.ac.jp

Competing interest: The authors declare that no competing interests exist.

**Abstract** Plasma membrane potential has been linked to cell proliferation for over 40 years in vertebrate cells. In this study, we experimentally demonstrated that membrane depolarization promotes mitosis and that this process depends on the voltage-dependent activation of extracellular signal-regulated kinase (ERK) in human cells. Notably, ERK activity showed a clear dependence on the membrane potential, independent of growth factor stimulation. This voltage dependence was observed even near the resting membrane potential, indicating that small shifts in the resting potential can influence proliferative activity. Voltage-dependent ERK activity is derived from the altered dynamics of phosphatidylserine and is not mediated by calcium influx from the extracellular space. These findings suggest that fundamental biological processes such as cell proliferation are regulated by the physicochemical properties of membrane lipids. This study highlights the broader physiological roles of membrane potentials beyond action potentials, which are well-established in neural systems.

## Introduction

Plasma membrane potential refers to the voltage difference across a cell's lipid bilayer, which is established by the asymmetric distribution and conductance of intracellular and extracellular ions. These ion gradients are maintained by the combined action of passive and active transport mechanisms involving various ion channels and transporters embedded in the membrane. Although each cell type exhibits a characteristic resting membrane potential, its physiological role has primarily been studied in excitable cells, such as muscle and nerve cells, whereas non-excitable cells have received comparatively little attention (*Hodgkin and Huxley, 1952*; *Neher and Sakmann, 1976*).

However, accumulating evidence suggests that the physiological relevance of membrane potentials extends beyond action potentials in excitable cells (*Zhou et al., 2015*). Membrane potential has been implicated in diverse biological processes including embryonic patterning, tissue regeneration, and cancer progression (*Levin, 2021*). It plays a critical role in establishing left–right asymmetry during early embryogenesis (*Levin et al., 2002*). Furthermore, the resting membrane potential

has been linked to a cell's proliferative capacity (*Sundelacruz et al., 2009*; *Blackiston et al., 2009*; *Cone and Tongier, 1973*; *Cone, 1971*; *Binggeli and Weinstein, 1986*), with depolarized cells consistently showing increased mitotic activity. In support of this view, *Voorhess et al., 1976* demonstrated that sustained depolarization promotes DNA synthesis in chick spinal cord neurons. These findings align with the observation that oncogenically transformed cells tend to be more depolarized than their normal counterparts (*Tokuoka and Morioka, 1957*). Despite numerous reports associating the membrane potential with cell proliferation, the molecular mechanisms underlying this relationship remain unclear.

Cell proliferation is tightly regulated by extracellular growth factors and their downstream intracellular signaling cascades, among which the mitogen-activated protein kinase (MAPK) pathway is central. A key component of this pathway, extracellular signal-regulated kinase (ERK), plays a crucial role in controlling cell proliferation (*Pagès, 1993*). ERK signaling is frequently upregulated in tumors but downregulated under contact inhibition or in growth-arrested cells (*Wayne et al., 2006*).

A recent study has shown that phosphatidylserine and K-Ras undergo changes in their nanoscale organization upon plasma membrane depolarization. Voltage-dependent ERK activation occurs when constitutively active K-RasG12V mutant is overexpressed (*Zhou et al., 2015*). However, the proliferative capacity was not the primary focus of this study, leaving the question of whether voltage-dependent ERK could explain membrane potential-dependent cell proliferation.

In this study, we provide experimental evidence that membrane depolarization enhances mitotic activity and that this effect is associated with voltage-dependent activation of ERK.

## Results

### Mitotic activity is enhanced by membrane depolarization via the MAPK cascade

Cell proliferation has long been associated with membrane voltage (*Cone, 1971*). We first examined whether a causal relationship existed between membrane potential and mitotic activity in our experimental system. To specifically monitor the mitotic phase of the cell cycle, U2OS cells were synchronized in the G1 phase by treatment with 2 mM thymidine for 20 h, followed by 8–9 h of normal incubation (G1 release). The cells were subjected to time-lapse imaging (*Figure 1A*). Cytokinesis, which was easily identified in bright-field images (*Figure 1B*), was quantified in each frame. Cells were maintained in serum-containing medium, except during the phase in which $K^+$ concentration was altered (*Figure 1A*).

The number of mitotic cells increased 14–15 h after G1 release, corresponding to approximately 6 h after the onset of time-lapse imaging (*Figure 1C*). Because only a limited number of conditions could be tested per experiment, the control condition (5K) was included for normalization. Mitotic activity, defined as the number of cells undergoing division, was significantly enhanced in cells exposed to 15 mM $K^+$ (*Figure 1C and D*). These results indicated that membrane depolarization promotes mitotic activity, supporting a causal relationship between membrane potential and proliferation. Notably, mitotic activity under $15K^+$ conditions was comparable to that observed under FBS-supplemented conditions (5K+FBS and 15K+FBS). The effects of depolarization and FBS were not additive, suggesting that the proliferative activity plateaued under both conditions.

Next, we investigated the potential involvement of ERK in depolarization-induced mitotic activity. Treatment with the MEK inhibitor U0126 abolished the proliferative effect of 15 mM $K^+$, indicating that MEK activation plays a pivotal role in depolarization-induced proliferation (*Figure 1C and D*).

Together, these findings suggest that MEK–ERK signaling mediates depolarization-induced mitotic activation. However, our results differ from those of a previous report showing that ERK activation occurs only when extracellular potassium exceeds 50 mM in cells expressing wild-type K-Ras (*Zhou et al., 2015*). Therefore, it remains unclear whether depolarization-induced proliferation is mediated by voltage-dependent ERK activation. To address this, we examined whether ERK could be activated within the physiological range of membrane depolarization.

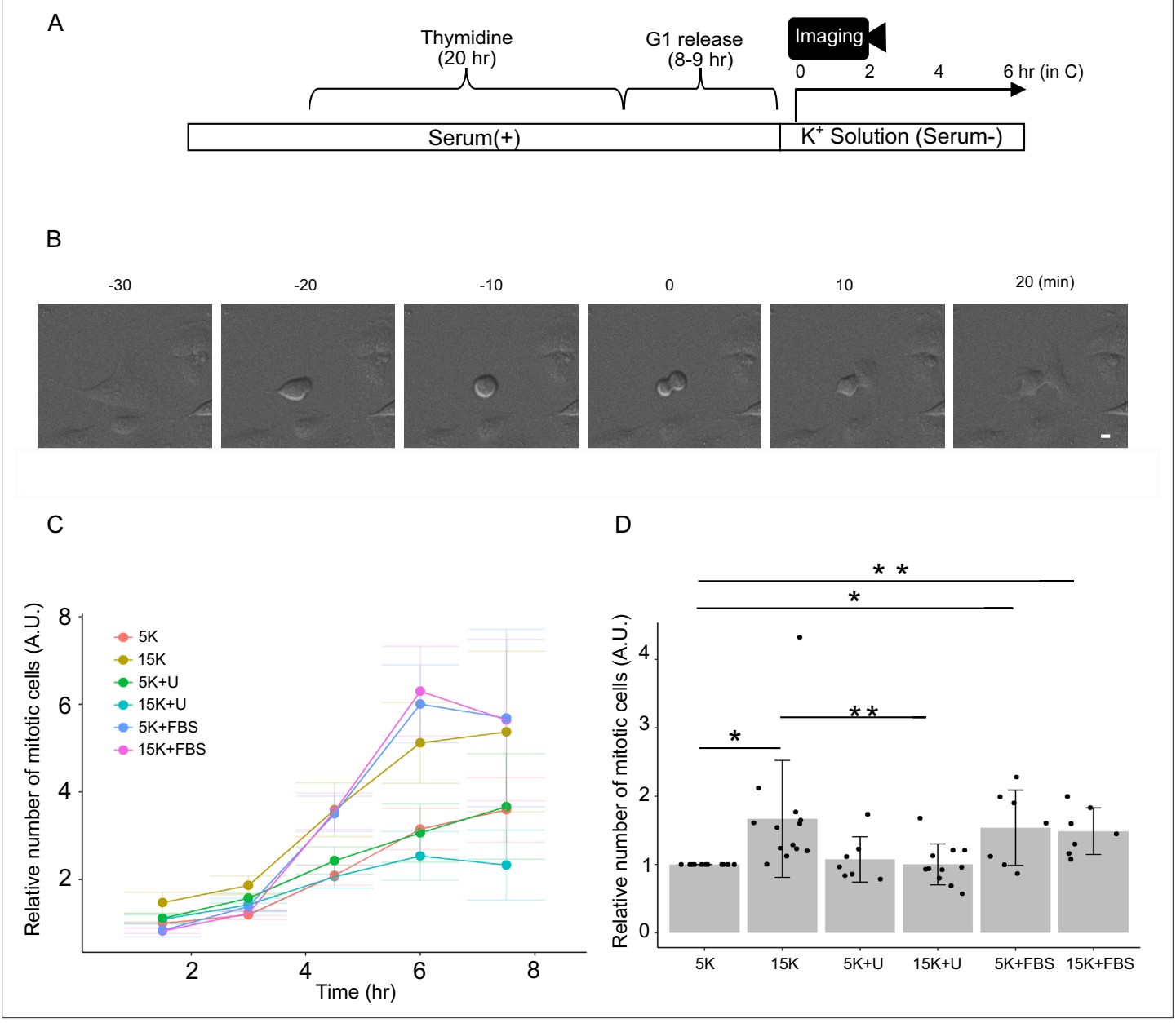

**Figure 1.** Mitotic activity is upregulated by membrane depolarization. (**A**) Experimental diagram. (**B**) Representative images showing morphological changes of a U2OS cell before, during, and after cell division. Time (min) indicated relative to the cytokinesis. Scale bar is 10 μm. (**C**) Mitotic cell counts were accumulated every 1.5 h, normalized by the number of mitotic cells in the first 1.5 h with 5K solution from the same experiment and plotted against time after the start of time-lapse imaging. U; U0126, FBS; 10% FBS. Data are mean ± SEM (**D**) Mitotic cell counts were accumulated for 6 h and normalized by the accumulated number of mitotic cells in 5K solution from the same experiment. 5K vs. 15K p=0.016 (one sample *t*-test), 5K vs. 5K+FBS p=0.041 (one sample *t*-test), 5K vs. 15K+FBS p=0.0094 (one sample *t*-test), 15K vs. 15K+U p=0.0040 (paired *t*-test). Data are from 13 (for 5K), 13 (for 15K), 7 (for 5K+U), 11 (for 15K+U), 7 (for 5K+FBS), and 7 (for 15K+FBS) independent experiments, respectively. Approximately 100–900 mitotic cells were observed within 6 h in each experiment (**C, D**).

## ERK activity has voltage dependency within the physiological range of the membrane potential

Although western blotting is a powerful tool for detecting all-or-nothing phenomena, it is not suitable for quantitatively capturing subtle changes or events with high cell-to-cell variability. Moreover, it is inadequate for analyzing detailed kinetics. To overcome these limitations, we used live-cell imaging. ERK activity was monitored by Förster resonance energy transfer (FRET) imaging using the ERK

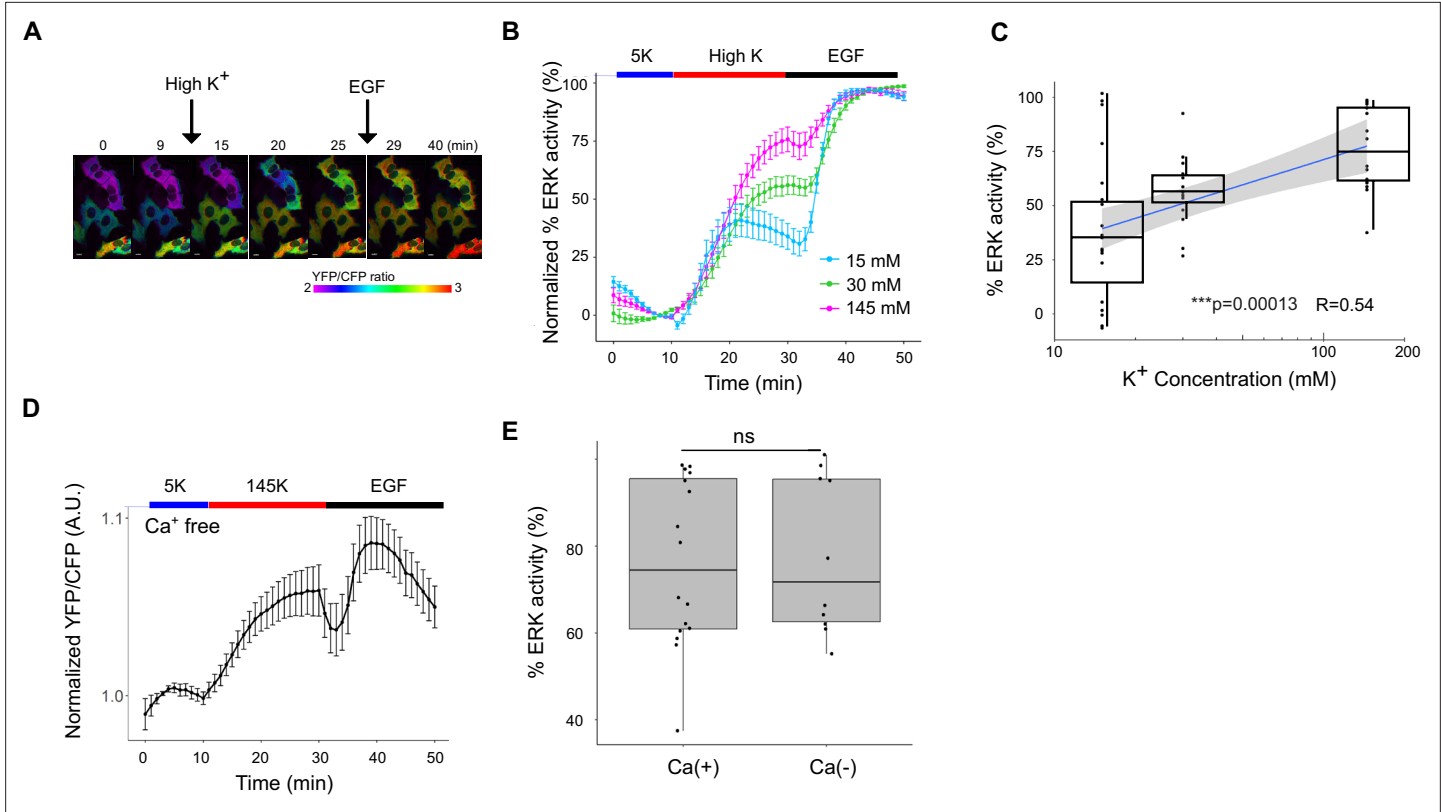

**Figure 2.** ERK is activated upon high K+perfusion. (**A**) YFP/CFP ratio images at the indicated time points. Serum-starved U2OS cells expressing EKAREV were perfused with 145 mM high K+ solution at 10 min, followed by the addition of 10 nM EGF at 30 min. Scale bar is 10 µm. The color scale indicates YFP/CFP ratio. (**B**) ERK activity with high K+ perfusion. Serum-starved cells were treated with various concentrations of K+ solution at 10 min, followed by the addition of 10 nM EGF at 30 min. Normalized % ERK (response by high K/response by EGF; see in the text) activity was calculated as follows: the normalized YFP/CFP (see in the text) was divided by the normalized maximum YFP/CFP after subtracting the mean normalized YFP/CFP at 8–10 min. Data are mean ± SEM. Data are from 16, 15, and 23 cells from 3, 4, and 6 independent experiments, for 145K+, 30K+, and 15K+, respectively. (**C**) The relationship between the logarithm of K+ concentration and the % ERK activity. R is Pearson's correlation coefficient. The gray area is 95% confidence interval. p=0.00013 (Kruskal–Wallis). N are the same as in (**B**). (**D**) Mean normalized YFP/CFP ratio perfused with $Ca^{2+}$-free 145 K+ high K solution. Starved cells were incubated with 5K+ ($Ca^{2+}$-free) solution, followed by perfusion with 145 mM K+ ($Ca^{2+}$-free) solution at 10 min. At 30 min, 10 nM EGF was added. Data are mean ± SEM. N=14 from two independent experiments. (**E**) The % ERK activity in the Ca(+) and Ca(-) solution. The data for Ca(+) were the same as in (**C**) (145K+). p=0.82 (Wilcoxon rank sum exact test) Data are from 16 cells from three independent experiments for Ca(-).

The online version of this article includes the following figure supplement(s) for figure 2:

**Figure supplement 1.** Non-normalized YFP/CFP ratio responded to high K+ perfusion.

**Figure supplement 2.** ERK was activated by high K+ perfusion in multiple cell types.

**Figure supplement 3.** ERK was not activated by perfusion nor hypotonic shock.

biosensor EKAREV, while the plasma membrane potential was manipulated by altering the extracellular potassium concentration. EKAREV is an intramolecular FRET biosensor that undergoes conformational changes depending on ERK activity (***Komatsu et al., 2011***). Ypet and ECFP were used as the FRET pair. For simplicity, we will refer to the pair as YFP and CFP, respectively.

When U2OS cells expressing EKAREV were perfused with 145 mM K+ solution, the YFP/CFP ratio increased over the subsequent 10–20 min, indicating ERK activation (***Figure 2A and B***). These kinetics are consistent with those reported previously (***Zhou et al., 2015***), confirming that our imaging system can capture depolarization-induced ERK activation.

For quantitative analysis, two types of normalization were combined: one to account for variation among dishes and the other for variation among cells. Since the raw YFP/CFP value varied among dishes (***Figure 2—figure supplement 1***), presumably due to the difference in the expression level of the EKAREV probe and the background noise, the YFP/CFP value before application of the high K+ solution was averaged and used as a reference. The ratio of the raw YFP/CFP value to the reference

value was defined as the normalized YFP/CFP value. To normalize variation among cells, cells were stimulated with EGF (10 ng/mL) at the end of the experiment, which presumably yielded a near-saturated YFP/CFP value (ERK activity). This value was used to determine the maximum ERK activity in each cell. We defined % ERK activity as the high K$^+$-induced ERK activity (relative to the reference) divided by the maximum activity with EGF stimulation. Quantitative analyses were performed on imaged single cells.

Perfusion with 145 mM K$^+$ strongly activated ERK (*Figure 2A and B*), and similar results were observed in HEK293T, HeLa, and A431 cells (*Figure 2—figure supplement 2*). The mean ERK activity with 145 mM high K$^+$ perfusion was unexpectedly close to 100% (*Figure 2B*), indicating that depolarization induced by 145 mM K$^+$ had a similar impact on cells as EGF stimulation. Because 145 mM extracellular K$^+$ is far beyond the physiological range, we next tested more moderate depolarizations. Contrary to a previous report (*Zhou et al., 2015*), our imaging-based system detected ERK activation with 30 mM K$^+$, and even with 15 mM K$^+$—the condition under which mitotic activity was enhanced. Thus, even mild depolarization near the physiological range is sufficient to activate ERK, which may promote cell division. Importantly, as U2OS cells express wild-type K-Ras rather than an oncogenic mutant (*Libert et al., 2024*), our results raise the possibility that voltage-dependent ERK activation may also occur in non-transformed cells. The average ERK activity with 30 mM K$^+$ perfusion was approximately half of the maximum. The average ERK activity was further reduced when perfused with 15 mM K$^+$ solution (*Figure 2B*). These results indicate that ERK has voltage-dependent activity not only at extreme depolarization, but also within near-physiological membrane potentials.

An additional advantage of our imaging system is its ability to analyze ERK activity at the single-cell level. When individual cells were examined for the responses, the overall distribution was not Gaussian, and a large proportion of cells perfused with 145 mM K$^+$ solution showed %ERK activity of 90% or higher (*Figure 2B and C*). On the other hand, % ERK activity of cells perfused with 30 mM K$^+$ solution was concentrated around 50–60%, and the average response was lower. The variance in response was larger in cells perfused with the 15 mM K$^+$ solution, with a significant proportion of cells showing no response (*Figure 2C*). The correlation coefficient between the % ERK activity and the logarithm of extracellular K$^+$ concentration was 0.54 (*Figure 2C*). Since membrane potential is proportional to the logarithm of extracellular K$^+$ (Nernst equation), these data confirm that ERK activity is voltage-dependent across both supraphysiological and physiological ranges.

ERK can be activated by multiple growth factors, as well as by stimuli such as neural excitation and mechanosensory input (*Harvey et al., 2008*; *Crozet and Levayer, 2023*), where extracellular Ca$^{2+}$ entry is sometimes implicated. For instance, ERK activation in neurons is abolished under Ca$^{2+}$-free conditions (*Corvol et al., 2005*), and mechanosensory stimulation via Piezo1-mediated Ca$^{2+}$ influx activates ERK. Therefore, we examined whether extracellular Ca$^{2+}$ entry was required for voltage-dependent ERK activation. Perfusion with Ca$^{2+}$-free 145 mM K$^+$ solution activated ERK to the same extent as EGF stimulation, indicating that extracellular Ca$^{2+}$ influx is not required (*Figure 2D and E*).

Previous studies have reported that ERK is activated in response to hypotonic cell swelling (*Schliess et al., 1995*; *Sadoshima et al., 1996*). However, under our experimental conditions, no ERK activation was observed at 70% or 50% osmolarity (*Figure 2—figure supplement 3*). We also confirmed that perfusion itself had negligible effects, as no ERK activation was detected upon the initiation of 5K$^+$ perfusion (*Figure 2—figure supplement 3*).

## ERK activity depends on plasma membrane potential

ERK was previously reported to be activated only above –10 mV with wild-type K-Ras (*Zhou et al., 2015*). To gain further insight into the relationship between voltage-dependent ERK activation and the absolute membrane potential, we employed electrophysiology in combination with live cell imaging.

ERK activity was monitored by EKAREV FRET imaging, whereas depolarization was achieved by the voltage-clamp electrophysiology. Because U2OS cells were not amenable to the giga-seal formation required for patch clamping, we used HEK293T cells stably expressing the ERK biosensor, EKAREV.

In a preliminary experiment, we performed whole-cell patch-clamp analysis, but ERK activation was not detected, even upon EGF stimulation, suggesting that the intracellular components necessary for ERK activation were depleted by the whole-cell configuration. Thus, we next turned to the perforated patch-clamp technique using gramicidine, which preserves intracellular components.

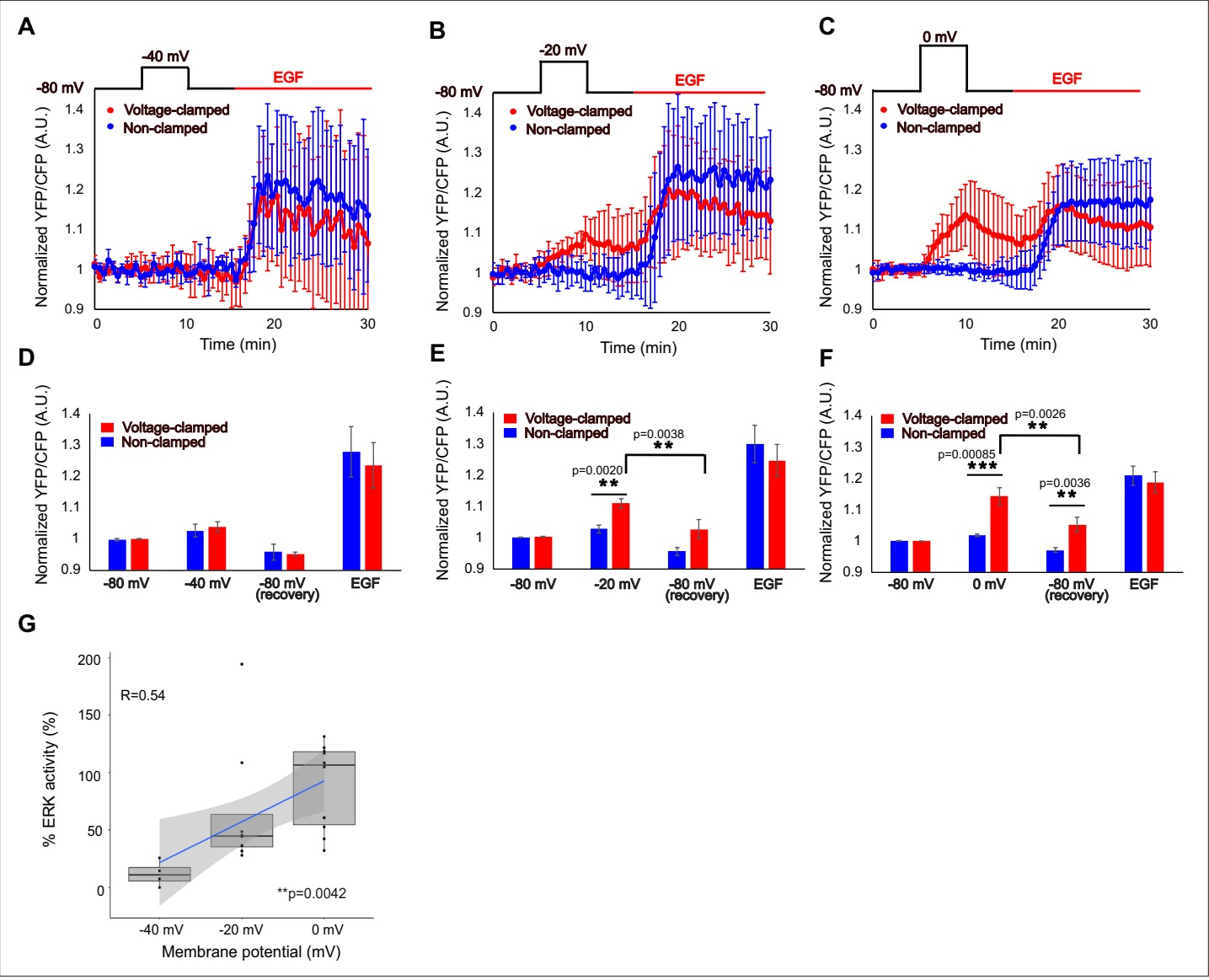

**Figure 3.** ERK activity depends on membrane voltage. (**A–C**) Mean normalized YFP/CFP ratio. Starved cells were depolarized from a holding potential of –80 mV to depolarized membrane potentials (–40, –20, and 0 mV for **A, B**, and **C,** respectively) at 5 min, followed by repolarization to –80 mV at 10 min. At 15 min, 10 nM EGF was perfused. Data are mean ± SD. N=4, 8, and 10 from two, five, and six independent experiments for –40, –20, and 0 mV, respectively. (**D–F**) Average normalized YFP/CFP ratio for the first 5 min (–80 mV), maximum normalized YFP/CFP ratio at depolarized phase (–40, –20, and 0 mV for **A, B**, and **C,** respectively), minimum normalized YFP/CFP ratio at repolarized phase (–80 mV) and maximum normalized YFP/CFP ratio at EGF stimulated phase (EGF), respectively. Data are mean ± SEM. The p–values were calculated using Student's *t*-test. (**G**) The relationship between membrane potential and % ERK activity. p=0.0042 (Kruskal–Wallis). R is Pearson's correlation coefficient. The gray area is 95% confidence interval.

HEK293T cells stably expressing EKAREV were depolarized from a holding potential of –80 mV. A control cell with the same optical view and whose membrane voltage was not manipulated is shown in blue (*Figure 3A–F*).

Depolarization to –40 mV did not upregulate ERK (*Figure 3A and D*). Depolarization from –80 mV to –20 mV significantly activated ERK (*Figure 3B and E*). Depolarization to 0 mV resulted in the strongest ERK activation, which approached the maximum level (*Figure 3C and F*).

In our experimental condition, HEK293T cells exhibited resting membrane potential ranging from –36 mV to –15 mV. This is in line with the lack of ERK activation at –40 mV.

Notably, ERK activity began to decline within 1 min after repolarization to –80 mV, a much faster timescale than the 30 min reported previously (*Zhou et al., 2015*). The rapid and reversible change

in ERK activity coupled with membrane potential contrasts with its activation by EGF, which usually remained sustained for more than 10 min (*Figure 3B, C, E and F*).

The correlation coefficient between % ERK activity and membrane potential was 0.54 (*Figure 3G*), suggesting that ERK activity was voltage-dependent, even near the resting membrane potential. We also confirmed that depolarization to 0 mV alone was sufficient to activate ERK to a level comparable (~88 ± 35%) to EGF stimulation (*Figure 3G*).

The variance of % ERK activity was larger at –20 mV compared to either 0 mV or –40 mV depolarization pulses (*Figure 3G*). We speculate that this variability arises from differences in the resting membrane potentials of individual cells, which were observed to range between –36 mV and –15 mV. At –20 mV, the extent of depolarization therefore depends on each cell's initial potential: in some cells, the shift produces little or no depolarization, whereas in others it results in substantial depolarization. This variability in the effective voltage change likely accounted for the broader distribution of ERK responses.

## MAPK cascade activation upon depolarization

We confirmed ERK activation upon high-$K^+$ perfusion by western blotting. In this experiment, we used 145 mM $K^+$ perfusion since 50 mM $K^+$ had previously been reported not to upregulate ERK activity by western blot (*Zhou et al., 2015*). U2OS cells were perfused with 145 mM $K^+$ solution and lysed in lysis buffer. Consistent with our imaging results, ERK was phosphorylated following high-concentration $K^+$ perfusion (*Figure 4A–B*).

Next, we examined whether the upstream components of the canonical MAPK signaling cascade were similarly affected under the same conditions. MEK and c-Raf, both located upstream of ERK, were also phosphorylated by 145 mM $K^+$ perfusion (*Figure 4A and B*). These findings indicate that membrane depolarization activates the MAPK cascade.

Our live-cell imaging data revealed that depolarization-induced ERK activation displays considerably slower kinetics than voltage-gated ion channels, which typically respond within a millisecond (*Bezanilla, 2000*). Therefore, we investigated the kinetics of upstream MAPK components using imaging-based approaches, which allowed for a more detailed temporal resolution.

## Ras was activated upon depolarization with rapid kinetics

Next, we examined the Ras activity upstream of c-Raf. Although Ras nanoclusters are known to activate the downstream MAPK cascade, there is no direct evidence that Ras itself is activated by depolarization. Ras activity was monitored using the FRET-based biosensor Raichu-Ras (*Mochizuki et al., 2001*). Raichu-Ras is an intramolecular biosensor that undergoes a conformational change upon Ras activation, resulting in increased FRET efficiency. When FRET occurs, the lifetime of the donor fluorophore decreases. To monitor Ras activity in the plasma membrane, we employed fluorescence lifetime imaging microscopy combined with FRET (FLIM-FRET) (*Yeung et al., 2008*) and confocal microscopy. U2OS cells expressing Raichu-Ras were perfused with a 145 mM $K^+$ solution, followed by stimulation with EGF (10 ng/mL).

The mean fluorescence lifetime of the donor fluorophore of Raichu-Ras at the plasma membrane decreased within 2 min after 145 mM $K^+$ perfusion, indicating Ras activation (*Figure 4C*). The activation kinetics were comparable between high $K^+$ stimulation and EGF stimulation, and were markedly faster than depolarization-induced ERK activation. These results are also consistent with previous reports (*Zhou et al., 2015*; *Murakoshi et al., 2004*). Taken together, these data suggested that Ras is activated by membrane depolarization with more rapid kinetics than ERK.

## The phosphatidylserine dynamics are involved in voltage-dependent ERK activity

Next, we investigated signaling events upstream of Ras. Changes in the membrane potential are expected to occur within milliseconds; thus, upstream events should occur with kinetics that are at least as fast as those of Ras. Plasma membrane depolarization induces the nanoscale reorganization of phosphatidylserine (*Zhou et al., 2015*). K-Ras, which targets the plasma membrane through electrostatic interactions with phosphatidylserine, undergoes nanoclustering and amplifies MAPK signaling in fibroblasts, neuroblastoma cells, and *Drosophila* neurons (*Zhou et al., 2015*). Based on these findings,

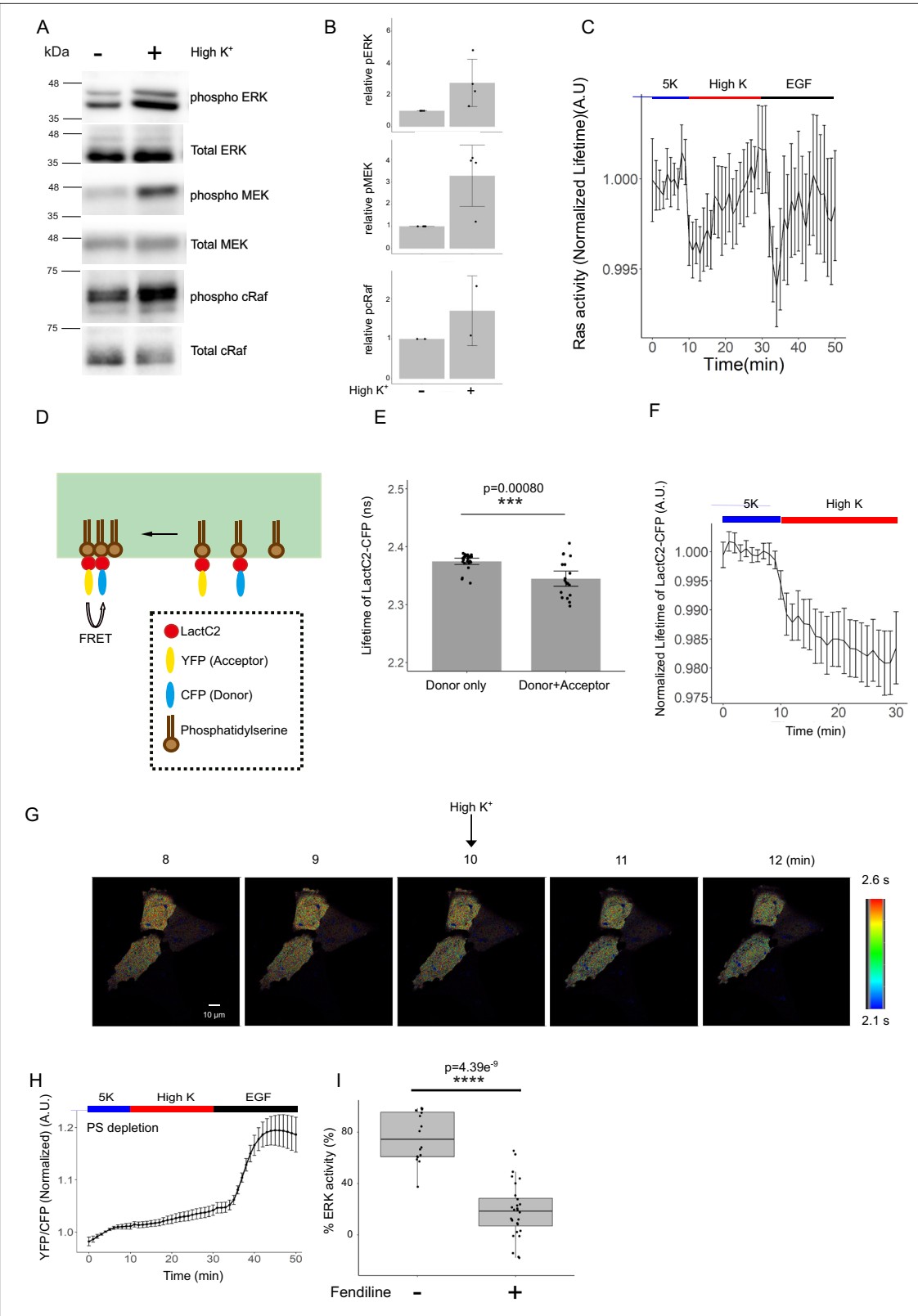

**Figure 4.** The molecular mechanism regulating depolarization-induced mitosis. (**A**) Immunoblot analysis of cells perfused with 145K$^+$ solution. (**B**) Quantification of relative mean band intensity of pERK, pMEK, and pcRaf from four, four and two independent experiments, respectively, conducted as in (**A**). Data are mean ± SD. (**C**) Ras activity was monitored by FLIM-FRET imaging. The donor (mTurquoise-GL) lifetime of U2OS cells expressing Raichu-Ras at the indicated time point. The experimental condition was the same as in *Figure 2A*. Data are mean ± SEM. N=17 from 4 independent

*Figure 4 continued on next page*

*Figure 4 continued*

experiments. (**D**) Schema of intermolecular FRET experiment. Phosphatidylserine clustering was monitored by FLIM-FRET imaging of U2OS cells co-expressing CFP-LactC2 and YFP-LactC2. (**E**) The fluorescence lifetime of CFP-LactC2. The fluorescence lifetime of CFP-LactC2 decreased when YFP-LactC2 was co-expressed in the same cells. p=0.00080 (Welch two-sample *t*-test) N=25 and 16 from six and five independent experiments, for CFP-LactC2 alone and co-expressing CFP-LactC2 and YFP-LactC2, respectively. (**F**) The normalized fluorescence lifetime of CFP-LactC2 was measured at the indicated time point. The experimental condition was the same as in *Figure 2A* except the EGF stimulation. Data are mean ± SEM. N=25 from six independent experiments. (**G**) CFP-LactC2 lifetime images at the indicated time points. The experimental condition is described above. Scale bar is 10 μm. The color scale indicates CFP-LactC2 lifetime. (**H**) Mean normalized YFP/CFP ratio from phosphatidylserine-depleted starved cells treated with 145 $K^+$ solution at 10 min, followed by the addition of 10 nM EGF. Data are mean ± SEM. N=27 from eight independent experiments. (**I**) The % ERK activity with or without Fendiline treatment. The data for Fendiline(-) were the same as in *Figure 2C* (145K+). p=4.4e-9 (Wilcoxon rank sum exact test).

The online version of this article includes the following source data for figure 4:

**Source data 1.** Original files for western blot analysis displayed in *Figure 4A*.

**Source data 2.** Files containing original western blots for *Figure 4A*, indicating the relevant bands and molecular weight marker.

we examined whether phosphatidylserine nanoclusters contribute to voltage-dependent ERK activation in U2OS cells.

Phosphatidylserine dynamics were monitored by intermolecular FRET using the phosphatidylserine-specific probe LactC2 (*Figure 4D*). The C2 domain of lactadherin (LactC2), which binds phosphatidylserine, enables visualization when fused with CFP or YFP (*Yeung et al., 2008*). The fluorescence lifetime of CFP-LactC2 was 2.375±0.014 ns, which decreased to 2.345±0.013 ns upon co-expression of YFP-LactC2 (*Figure 4E*), indicating that phosphatidylserine nanoclustering could be monitored by FLIM-FRET. Importantly, the fluorescence lifetime of CFP-LactC2 in cells co-expressing YFP-LactC2 decreased within 2 min of 145 mM $K^+$ perfusion, demonstrating depolarization-induced phosphatidylserine nanoclustering (*Figure 4F and G*). The kinetics of nanoclustering are comparable to those of Ras activation, which is consistent with the notion that phosphatidylserine and Ras interact.

To determine whether phosphatidylserine dynamics were required for voltage-dependent ERK activation, phosphatidylserine was depleted by treating cells with fendiline for 48 h (*Cho et al., 2016*). Under these conditions, high $K^+$-induced ERK activation was markedly reduced (*Figure 4H and I*). These findings indicated that phosphatidylserine dynamics are critical for voltage-dependent ERK activation in U2OS cells.

## Discussion

The relationship between cell proliferation and the membrane potential has been widely discussed since the 1970s. However, as few studies have addressed the molecular mechanisms involved, the relationship between membrane potential and proliferative capacity remains unclear. Regarding the molecular mechanisms involved, a study conducted in 2015 demonstrated a correlation between MAP activity and membrane potential. However, ERK voltage dependence within the physiological membrane potential range has not yet been demonstrated. Therefore, it remains unclear whether voltage-dependent MAP activation can explain the link between membrane potential and cell proliferation. In this study, we experimentally demonstrated a causal relationship between the membrane potential and mitotic activity (cytokinesis) within the physiological membrane potential range. This effect is associated with depolarization-induced ERK activation via phosphatidylserine dynamics. Thus, we were able to clarify the molecular mechanism of a phenomenon that has long been suggested to be a correlation between membrane potential and proliferative capacity.

We showed that membrane depolarization induces mitotic activity through ERK activation (*Figure 1*), indicating that membrane potential is not merely correlated with, but can directly regulate mitotic activity. Some reports have indicated that ERK plays an important role in S phase entry (*Albeck et al., 2013*), whereas others have proposed a role in mitosis (*Shapiro et al., 1998*; *Willard and Crouch, 2001*; *Iwamoto et al., 2016*). Further studies are needed to elucidate which step(s) of the cell cycle are regulated by the membrane potential.

A potential limitation of extracellular $K^+$-based approaches is their reliance on the Nernst equation to estimate membrane potential, which oversimplifies the actual situation by neglecting voltage-gated ion channel activity and compensatory mechanisms. To directly address this concern, we measured membrane potential using the perforated patch-clamp technique and confirmed that the potential

was stable during perfusion with 145 mM K$^+$ (only a 1–5 mV drift within 20 min). Moreover, we used a voltage clamp to precisely control the membrane potential and demonstrated that ERK activity was directly regulated by the voltage itself, excluding the influence of other secondary factors. An additional strength of electrophysiology is its ability to examine the effects of repolarization, which is difficult to assess with conventional perfusion-based methods owing to slow solution exchange.

A key feature of the ERK response to voltage, compared to EGF stimulation, was the graded relationship between the stimulus and the response (*Figures 2 and 3*). Notably, ERK activity is down-regulated upon membrane repolarization, but remains active for at least 5 min after growth factor binding (*Kiyatkin et al., 2020*). This suggests that the membrane potential acts as a fine tuner of ERK, in contrast to growth factor stimulation, which functions more like an on/off switch.

Using FRET-based imaging, we analyzed the kinetics of each step in the signaling pathway. Nano-clustering of phosphatidylserine and subsequent Ras activation occurred within 2 min (*Figure 4C and F*), whereas ERK activation required more than 5 min (*Figure 2B*). This delay is reasonable given that several upstream molecules must be activated before ERK.

Another advantage of imaging-based analysis is the ability to quantitatively measure ERK activity in individual cells. Surprisingly, we found large cell-to-cell variability in depolarization-induced ERK activation. Importantly, such variability is averaged out and is thus invisible in western blot analysis, which is commonly used to study ERK signaling. Surprisingly, depolarization to 0 mV activated ERK to the same extent as EGF stimulation.

Coupled with electrophysiology, imaging also allows us to monitor ERK activity with a temporal resolution that perfusion or biochemical assays cannot achieve. ERK was activated within 1 min of depolarization and remained elevated for at least 5 min. Deactivation kinetics upon repolarization were similar to those upon activation (*Figure 3*).

We also demonstrated that ERK activity exhibits a clear voltage dependence, spanning from the resting membrane potential to states of strong depolarization. This observation is consistent with the idea that voltage-dependent ERK activation arises from changes in phosphatidylserine dynamics in the plasma membrane, which are governed by the fundamental physicochemical properties of the lipid bilayer. Taken together, our findings indicate that plasma membrane potential should be regarded as a previously unrecognized but important regulator of ERK signaling.

In this study, we primarily used U2OS cells because their flat morphology makes them suitable for live-cell FRET imaging. Although cancer cell lines, including U2OS, may display bioelectric properties that differ from those of noncancerous cells, our findings raise the possibility that voltage-dependent ERK activation is a fundamental and broadly applicable phenomenon rather than a feature specific to osteosarcoma cells. This conclusion is supported by the fact that essential components of this pathway, namely phosphatidylserine, Ras, and MAPK (including ERK), are ubiquitously expressed in mammalian cells. Consistent with this finding, we observed voltage-dependent ERK activation across multiple cell lines: U2OS, HeLa, HEK293T, and A431 cells (*Figure 2—figure supplement 2*). These observations indicate that the mechanism we describe is not cell-type-restricted, but rather a universal property of mammalian cells.

Zhou et al. reported that phosphatidylserine and PIP$_2$ undergo nanoscale reorganization upon plasma membrane depolarization (*Zhou et al., 2015*). K-Ras also changes its nanoscale organization through electrostatic interactions between its basic residues and anionic phosphatidylserine (*Zhou et al., 2015*; *Zhou and Hancock, 2015*). Such interactions may also occur between clusters of basic amino acids in other proteins and anionic phospholipids such as phosphatidylserine or PIP$_2$. For example, a cluster of basic residues in EGFR interacts with PIP$_2$, and clustering of EGFR leads to its activation (*Wang et al., 2014*; *Abd Halim et al., 2015*). This suggests that EGFR activity may also be regulated by the membrane potential. Several other proteins interact with PIP$_2$ and contain clusters of basic residues, suggesting that the membrane potential regulates a broad range of proteins and, thereby, many physiological processes.

We showed that the membrane potential modulated phosphatidylserine dynamics (*Figure 4F and G*). The relationship between electrostatic potential and membrane organization has been discussed previously (*Malinsky et al., 2016*). Transmembrane potential affects the lateral sorting of membrane components, as well as the positioning of polar lipid headgroups (*Malinsky et al., 2016*). In addition, membrane lipid viscosity, which depends on lipid/protein composition and organization, can be altered by voltage (*O'Shea et al., 1984*; *Herman et al., 2004*; *Grossmann et al., 2007*). However, the

precise mechanism through which transmembrane potentials modulate lipid and protein organization remains unclear.

Overall, our results link three elements—plasma membrane voltage, ERK activity, and cell proliferation—and propose a new signaling cascade that regulates cell division. These findings suggest that ion channels and transporters that regulate membrane potential are potential therapeutic targets for cancer. Moreover, because ERK is involved in differentiation, migration, senescence, and apoptosis (*Sun et al., 2015*), the membrane potential may also regulate these processes, representing additional opportunities for drug development.

What is the biological significance of membrane potential regulation in cell proliferation? The stochastic ERK activation reported by *Aoki et al., 2013* may result from fluctuations in membrane potential. Furthermore, intercellular propagation of membrane potential via gap junctions can increase ERK activity and synchronize proliferation (*Aoki et al., 2013*). Such synchronized proliferation could be advantageous in tissues such as the epithelium. Our finding that membrane potential acts as an analog modulator of ERK activity is consistent with this hypothesis and may enable the nuanced regulation of cell proliferation.

It is now evident that the physiological significance of membrane potential extends beyond ion channel function. Given that the membrane potential is a physical property that influences all cellular components, including proteins and lipids, many additional biological phenomena regulated by voltage remain to be explored.

## Materials and methods
### Plasmid and reagents
The expression vectors for the EKAREV and Raichu-Ras plasmids were kindly provided by Dr. Michiyuki Matsuda (Kyoto University). mRFP-Lact-C2 was a gift from Sergio Grinstein (Addgene plasmid # 74061; https://www.addgene.org/74061/; Addgene_74061; *Yeung et al., 2008*). YFP-LactC2 and CFP-LactC2 were generated from the mRFP-LactC2 plasmid by exchanging mRFP with YPet and SECFP, respectively. Recombinant human EGF was purchased from Thermo Fisher Scientific. Fendiline was purchased from CAYMAN Chemicals. Gramicidin was purchased from Sigma-Aldrich. U0126 was purchased from Promega.

### Cells
U2OS cells were obtained from the European Collection of Authenticated Cell Cultures (ECACC), and HEK293T, A431, and Hela cells were purchased from the RIKEN Cell Bank (Tsukuba, Japan). HEK293T cells were used because they are widely employed for electrophysiological experiments and allow for high levels of probe expression, which was required in this study. All cell lines were authenticated using short tandem repeat (STR) profiling and tested negative for mycoplasma contamination. U2OS cells were maintained in McCoy's 5 A medium supplemented with 10% fetal bovine serum. HEK293T cells stably expressing EKAREV were generated by PiggyBac transposon system transfection with EKAREV/pPBbsr2 and mPBase/pCMV vectors and were maintained in Dulbecco's Modified Eagle Medium (DMEM) supplemented with 10% fetal bovine serum (FBS). Cells were cultured in a 5% $CO_2$-humidified environment at 37°C.

### Time-lapse imaging with high $K^+$ perfusion
U2OS cells were plated on a glass-based dish (Matsunami) and transfected with the EKAREV plasmid using the TransIT-LT1 reagent (Mirus) according to the manufacturer's instructions. The cells were serum-starved for 3–16 h before time-lapse FRET imaging. The cells were imaged using an inverted microscope (Ti2-E; Nikon) equipped with a 60× objective (NA = 1.4; Nikon), a CMOS camera (Zyla4.2; Andor), and an LED illumination system (xCite Xylis; Excelitas Technologies). A W-VIEW GEMINI image-splitting system (Hamamatsu Photonics) was used to acquire the CFP and YFP images simultaneously. The following filters and dichroic mirrors were used: 439/24 for excitation; 483/32 (CFP) and 542/27 (YFP) for emission; and FF509-fDi01 for the dichroic mirror. FLIM-FRET images were acquired using a Stellaris (Leica) confocal microscope with excitation at 448 nm. The fluorescence lifetime of the donor fluorophore was measured at 460–500 nm with a pinhole size of 1 AU, so that only signals from the plasma membrane were detected. The perfusion solution was changed using a valve-controlled

gravity perfusion system (VC3-4PG) controlled by an Arduino. The solutions used were as follows (in mM), 5K; 140 NaCl, 5 KCl, 1 CaCl$_2$, and 1 MgCl$_2$. 15K; 130 NaCl, 15 KCl, 1 CaCl$_2$, 1 MgCl$_2$. 30K; 115 NaCl, 30 KCl, 1 CaCl$_2$, 1 MgCl$_2$. 145K; 145 KCl, 1 CaCl$_2$, 1 MgCl$_2$. 5K (Ca free); 140 NaCl, 5 KCl, 1 EGTA, 1 MgCl$_2$. 145K (Ca free); 145 KCl, 1 EGTA 1 MgCl$_2$. All solutions contained 20 mM HEPES (pH 7.4). The osmolarity was adjusted to 296–305 mOsm/L using glucose. The EGF solution was prepared using a low-K $^+$solution (5 mM K$^+$).

## Patch clamping

HEK293T cells stably expressing EKAREV were plated on a glass-based dish (Matsunami) and serum-starved for 16 h before the start of the experiment. The gramicidin-perforated patch-clamp experiment was performed using a Multiclamp 700 B amplifier (Digidata 1440A) and pClamp software (Axon). Gramicidin (60 μg/mL) in a patch solution (145 mM KCl, 5 mM EGTA, 10 mM HEPES pH 7.4) was sonicated for 5 min and backfilled into the patch pipette. The 5 K solution was used as the external patch solution. FRET images were acquired using a confocal microscope LSM510Meta (Zeiss) excited at 458 nm. The filters used for CFP and YFP were 475–525 and LP530, respectively. We set the potential to –80 mV immediately after the giga-seal formation and waited for at least 5 min to allow pore formation by gramicidin. We started imaging only after membrane potential was expected to have reached a steady state at –80 mV.

## Imaging analysis

Dead and unhealthy cells that did not respond to EGF stimulation were excluded from the analysis. The average intensity of multiple background ROIs was used as the background and subtracted from each image. The fluorescence intensities of the YFP and CFP channels were averaged for each cell, and the results were exported as a CSV file. YFP/CFP values were calculated using Excel and R Studio for the patch-clamp and high-K$^+$ perfusion experiments, respectively. The % ERK activity was calculated by dividing the maximum normalized YFP/CFP ratio after high K$^+$ perfusion or depolarization by the maximum normalized YFP/CFP ratio after EGR perfusion for each cell. For FLIM-FRET experiments, the average fluorescence lifetime was calculated for each cell. The average fluorescence lifetime of the first 10 min was used as a reference. The normalized fluorescence lifetime was defined as the ratio of the fluorescence lifetime to the reference value. All image analyses were performed using ImageJ software.

## Western blot

The following antibodies were used for western blotting: total ERK (SC514302; Santa Cruz Biotechnology), phospho-ERK (4370; Cell Signaling Technology), total MEK (4394; Cell Signaling Technology), phospho-MEK (2338; Cell Signaling Technology), total c-Raf (9422; Cell Signaling Technology), phospho-c-Raf (CST9431). Cells were lysed in 50 mM Tris (pH 7.5), 150 mM NaCl, 1 mM EDTA, 1% NP40 with protease and phosphatase inhibitor cocktail (Thermo Scientific), 1 mM sodium orthovanadate. Protein samples (10 μg per lane) were separated by SDS-PAGE, immunoblotted onto polyvinylidene fluoride (PVDF) membranes, and detected by ECL using a charge-coupled device (CCD) camera. The same protein samples were probed with multiple antibodies. To achieve this, the membranes were cut horizontally after protein transfer, and each portion was blotted with a different antibody. This approach was necessary because the total amount of protein available was extremely limited, as the lysates were prepared under the same experimental conditions as those used for imaging. Quantification was performed using ImageJ software.

## Mitosis analysis

U2OS cells were synchronized in the G1 phase by treatment with 2 mM thymidine for 20 h. Time-lapse imaging was conducted 8–9 h after G1 release using an inverted confocal microscope (Stellaris, Leica) with a 20× objective (NA = 0.75) in DIC mode. Images were obtained every 10 min in solutions with different K$^+$ concentrations without serum (*Figure 1A*). The number of mitotic cells was counted in each frame. The FBS condition included 10% FBS.

## Statistical analysis

Statistical analyses were performed using RStudio. The level of statistical significance is indicated by asterisks: ns, $p > 0.05$; *$p < 0.05$; **$p < 0.01$; ***$p < 0.001$; ****$p < 0.0001$.

## Acknowledgements

We thank Dr. Michiyuki Matsuda (Kyoto University) for providing the EKAREV and RaichuEV-HRas plasmids. We also thank Dr. Michiyuki Matsuda, Dr. Tohru Ishitani, and Dr. Jianmin Cui for critically reading this manuscript. KAKENHI 20K07268 to MS Naito Foundation to MS Takeda Science Foundation to MS.

## Additional information

### Funding

| Funder | Grant reference number | Author |
|---|---|---|
| KAKENHI | 20K07268 | Mari Sasaki |
| Naito Foundation | | Mari Sasaki |
| Takeda Science Foundation | | Mari Sasaki |

The funders had no role in study design, data collection and interpretation, or the decision to submit the work for publication.

### Author contributions

Mari Sasaki, Conceptualization, Data curation, Software, Formal analysis, Funding acquisition, Investigation, Methodology, Writing – original draft, Project administration, Writing – review and editing; Masanobu Nakahara, Software, Formal analysis; Takuya Hashiguchi, Investigation; Fumihito Ono, Writing – review and editing

### Author ORCIDs

Mari Sasaki https://orcid.org/0009-0006-6582-1805
Fumihito Ono https://orcid.org/0000-0001-7532-5262

Reviewer #1 (Public review): https://doi.org/10.7554/eLife.101613.4.sa1
Reviewer #2 (Public review): https://doi.org/10.7554/eLife.101613.4.sa2
Reviewer #3 (Public review): https://doi.org/10.7554/eLife.101613.4.sa3
Author response https://doi.org/10.7554/eLife.101613.4.sa4

## Additional files

### Supplementary files
MDAR checklist

### Data availability
All data generated or analyzed during this study are included in this article and its source data files. Figure 4-source data 1 and Figure 4-source data 2 provide the uncropped blots.

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
