## [Editor Report · eLife Assessment]

This **important** article employs multiple experimental approaches and presents evidence that changes in membrane voltage directly affect ERK signaling to regulate cell division. This result is relevant because it supports an ion channel-independent pathway by which changes in membrane voltage can affect cell growth. The evidence now presented is **solid** and the data support the conclusions. This article should be of interest to a broad readership in the areas of cell and developmental biology and electrophysiology.

---

## [Referee Report · Reviewer #1 (Public review)]

This is a contribution to the field of developmental bioelectricity. How do changes of resting potential at the cell membrane affect downstream processes? Zhou et al. reported in 2015 that phosphatidylserine and K-Ras cluster upon plasma membrane depolarization and that voltage-dependent ERK activation occurs when constitutively active K-RasG12V mutants are overexpressed. In this paper, the authors advance the knowledge of this phenomenon by showing that membrane depolarization up-regulates mitosis and that this process is dependent on voltage-dependent activation of ERK. ERK activity's voltage-dependence is derived from changes in the dynamics of phosphatidylserine in the plasma membrane and not by extracellular calcium dynamics. This paper reports an interesting and important finding. It is somewhat derivative of Zhou et al., 2015 (https://www.science.org/doi/full/10.1126/science.aaa5619). The main novelty seems to be that they find quantitatively different conclusions upon conducting similar experiments, albeit with a different cell line (U2OS) than those used by Zhou et al. Sasaki et al. do show that increased K+ levels increase proliferation, which Zhou et al. did not look at. The data presented in this paper are a useful contribution to a field often lacking such data.

---

## [Referee Report · Reviewer #2 (Public review)]

Sasaki et al. use a combination of live-cell biosensors and patch-clamp electrophysiology to investigate the effect of membrane potential on the ERK MAPK signaling pathway, and probe associated effects on proliferation. This is an effect that has long been proposed, but a convincing demonstration has remained elusive, because it is difficult to perturb membrane potential without disturbing other aspects of cell physiology in complex ways. The time-resolved measurements here are a nice contribution to this question, and the perforated patch clamp experiments with an ERK biosensor are fantastic - they come closer to addressing the above difficulty of perturbing voltage than any prior work. It would have been difficult to obtain these observations with any other combination of tools.

Comments on previous revisions:

The authors have done a good job addressing the comments on the previous submission.

---

## [Referee Report · Reviewer #3 (Public review)]

Summary:

This paper demonstrates that membrane depolarization induces a small increase in cell entry into mitosis. Based on previous work from another lab, the authors propose that ERK activation might be involved. They show convincingly using a combination of assays that ERK is activated by membrane depolarization. They show this is Ca2+ independent and is a result of activation of the whole K-Ras/ERK cascade which results from changed dynamics of phosphatidylserine in the plasma membrane that activates K-Ras. Although the activation of the Ras/ERK pathway by membrane depolarization is not new, linking it to an increase in cell proliferation is novel.

Strengths:

A major strength of the study is the use of different techniques - live imaging with ERK reporters, as well as Western blotting to demonstrate ERK activation as well as different methods for inducing membrane depolarization. They also use a number of different cell lines. Via Western blotting the authors are also able to show that the whole MAPK cascade is activated.

Weaknesses:

In the previous round of revisions, the authors addressed the issues with Figure 1, and the data presented are much clearer. The authors did also attempt to pinpoint when in the cell cycle ERK is having its activity, but unfortunately, this was not conclusive.

---

## [Author Response]

The following is the authors’ response to the previous reviews

**Reviewer #1 (Public review):**
Summary:This is a contribution to the field of developmental bioelectricity. How do changes of resting potential at the cell membrane affect downstream processes? Zhou et al. reported in 2015 that phosphatidylserine and K-Ras cluster upon plasma membrane depolarization and that voltage-dependent ERK activation occurs when constitutively active K-RasG12V mutants are overexpressed. In this paper, the authors advance the knowledge of this phenomenon by showing that membrane depolarization up-regulates mitosis and that this process is dependent on voltage-dependent activation of ERK. ERK activity's voltage-dependence is derived from changes in the dynamics of phosphatidylserine in the plasma membrane and not by extracellular calcium dynamics. This paper reports an interesting and important finding. It is somewhat derivative of Zhou et al., 2015. (https://www.science.org/doi/full/10.1126/science.aaa5619). The main novelty seems to be that they find quantitatively different conclusions upon conducting similar experiments, albeit with a different cell line (U2OS) than those used by Zhou et al. Sasaki et al. do show that increased K+ levels increase proliferation, which Zhou et al. did not look at. The data presented in this paper are a useful contribution to a field often lacking such data.Strengths:Bioelectricity is an important field for areas of cell, developmental, and evolutionary biology, as well as for biomedicine. Confirmation of ERK as a transduction mechanism and a characterization of the molecular details involved in the control of cell proliferation are interesting and impactful.Weaknesses:The authors lean heavily on the assumption that the Nernst equation is an accurate predictor of membrane potential based on K+ level. This is a large oversimplification that undermines the author's conclusions, most glaringly in Figure 2C. The author's conclusions should be weakened to reflect that the activity of voltage gated ion channels and homeostatic compensation are unaccounted for.

We appreciate the reviewer’s thoughtful comment regarding our reliance on the Nernst equation to estimate membrane potential. We agree that the Nernst equation is a simplification and does not account for the activity of other ions, voltage-gated channels, or homeostatic compensation mechanisms. To address this concern, we conducted electrophysiological experiments in which the membrane potential was directly controlled using the perforated patch-clamp technique (Fig. 3). Under these conditions, we also monitored the membrane potential and confirmed that there was negligible drift within 20 minutes of perfusion with 145 mM K^⁺^ (only a 1–5 mV change). These results suggest that the influence of voltage-gated channels and homeostatic compensation is minimal in our experimental setup. We revised the manuscript to clarify these limitations and to present our conclusions more cautiously in light of this point.

“A potential limitation of extracellular K^⁺^-based approaches is their reliance on the Nernst equation to estimate membrane potential, which oversimplifies the actual situation by neglecting voltage-gated ion channel activity and compensatory mechanisms. To directly address this concern, we measured membrane potential using the perforated patch-clamp technique and confirmed that the potential was stable during perfusion with 145 mM K^⁺^ (only a 1–5 mV drift within 20 min). Moreover, we used a voltage clamp to precisely control the membrane potential and demonstrated that ERK activity was directly regulated by the voltage itself, excluding the influence of other secondary factors. An additional strength of electrophysiology is its ability to examine the effects of repolarization, which is difficult to assess with conventional perfusion-based methods owing to slow solution exchange.”

There are grammatical tense errors are made throughout the paper (ex line 99 "This kinetics should be these kinetics")

We thank the reviewer for pointing out the grammatical errors. We carefully revised the entire manuscript.

Line 71: Zhou et al. use BHK, N2A, PSA-3 cells, this paper uses U2OS (osteosarcoma) cells. Could that explain the differences in bioelectric properties that they describe? In general, there should be more discussion of the choice of cell line. Why were U2OS cells chosen? What are the implications of the fact that these are cancer cells, and bone cancer cells in particular? Does this paper provide specific insights for bone cancers? And crucially, how applicable are findings from these cells to other contexts?

We thank the reviewer for this valuable comment regarding the choice of cell line. We selected U2OS cells primarily because they are well suited for live-cell FRET imaging. We did not use BHK, N2A, or PSA-3 cells, and therefore it is difficult for us to provide a clear comparison with the specific bioelectric properties reported in Zhou et al. Nevertheless, we agree that cancer cell lines, including U2OS, may exhibit bioelectric properties that differ from those of non-cancerous cells. While this could be a potential limitation, we are inclined to consider voltage-dependent ERK activation to be a fundamental and generalizable phenomenon, not restricted to osteosarcoma cells. The key components of this pathway—phosphatidylserine, Ras, MAPK (including ERK)—are expressed in essentially all mammalian cells. In support of our view, we observed voltage-dependent ERK activation not only in U2OS cells but also in HeLa, HEK293, and A431 cells. These results strongly suggest that the mechanism we describe is not cell-type specific but rather a universal feature of mammalian cells. In the revised Discussion, we expanded our rationale to choose U2OS cells, while addressing the potential implications of using a cancer-derived cell line.

“In this study, we primarily used U2OS cells because their flat morphology makes them suitable for live-cell FRET imaging. Although cancer cell lines, including U2OS, may display bioelectric properties that differ from those of noncancerous cells, our findings raise the possibility that voltage-dependent ERK activation is a fundamental and broadly applicable phenomenon rather than a feature specific to osteosarcoma cells. This conclusion is supported by the fact that essential components of this pathway, namely phosphatidylserine, Ras, and MAPK (including ERK), are ubiquitously expressed in mammalian cells. Consistent with this finding, we observed voltage-dependent ERK activation across multiple cell lines: U2OS, HeLa, HEK293, and A431 cells (Fig.S2). These observations indicate that the mechanism we describe is not cell-type-restricted, but rather a universal property of mammalian cells.”

Line 115: The authors use EGF to calibrate 'maximal' ERK stimulation. Is this level near saturation? Either way is fine, but it would be useful to clarify.

We thank the reviewer for raising this important point. The YFP/CFP ratio obtained after EGF stimulation is generally considered to represent saturation levels detectable by EKAREV imaging. However, we acknowledge that it remains uncertain whether 10 ng/mL EGF induces the absolute maximal ERK activity in all contexts. To clarify this point, we revised the manuscript (result) text as follows:

“To normalize variation among cells, cells were stimulated with EGF (10 ng/mL) at the end of the experiment, which presumably yielded a near-saturated YFP/CFP value (ERK activity). This value was used to determine the maximum ERK activity in each cell”

Line 121: Starting line 121 the authors say "Of note, U2OS cells expressed wild-type K-Ras but not an active mutant of K-Ras, which means voltage dependent ERK activation occurs not only in tumor cells but also in normal cells". Given that U2OS cells are bone sarcoma cells, is it appropriate to refer to these as 'normal' cells in contrast to 'tumor' cells?

We thank the reviewer for pointing this out. We agree that it is not appropriate to contrast U2OS cells with “normal” cells, since they are sarcoma-derived. To address this point, we revised the sentence to weaken the claim and avoid the misleading terminology.

“Importantly, as U2OS cells express wild-type K-Ras rather than an oncogenic mutant (16), our results raise the possibility that voltage-dependent ERK activation may also occur in non-transformed cells.”

Line 101: These normalizations seem reasonable, the conclusions sufficiently supported and the requisite assumptions clearly presented. Because the dish-to-dish and cell-to-cell variation may reflect biologically relevant phenomena it would be ideal if non-normalized data could be added in supplemental data where feasible.

We thank the reviewer for this helpful suggestion. As recommended, we added representative non-normalized data in the Supplemental Figure S1, which illustrates the non-normalized variation across cells and dishes.

Figure 2C is listed as Figure 2D in the textThere is no Figure 2F (Referenced in line 148)

We thank the reviewer for pointing out these errors. The incorrect figure citations were corrected.

**Reviewer #2 (Public review):**
Sasaki et al. use a combination of live-cell biosensors and patch-clamp electrophysiology to investigate the effect of membrane potential on the ERK MAPK signaling pathway, and probe associated effects on proliferation. This is an effect that has long been proposed, but a convincing demonstration has remained elusive, because it is difficult to perturb membrane potential without disturbing other aspects of cell physiology in complex ways. The time-resolved measurements here are a nice contribution to this question, and the perforated patch clamp experiments with an ERK biosensor are fantastic - they come closer to addressing the above difficulty of perturbing voltage than any prior work. It would have been difficult to obtain these observations with any other combination of tools.However, there are still some concerns as detailed in specific comments below:Specific comments:(1) All the observations of ERK activation, by both high extracellular K+ and voltage clamp, could be explained by cell volume increase (more discussion in subsequent comments). There is a substantial literature on ERK activation by hypotonic cell swelling (e.g. https://doi.org/10.1042/bj3090013, https://doi.org/10.1002/j.1460-2075.1996.tb00938.x, among others). Here are some possible observations that could demonstrate that ERK activation by volume change is distinct from the effects reported here:(i) Does hypotonic shock activate ERK in U2OS cells?(ii) Can hypotonic shock activate ERK even after PS depletion, whereas extracellular K+ cannot?(iii) Does high extracellular K+ change cell volume in U2OS cells, measured via an accurate method such as fluorescence exclusion microscopy?(iv) It would be helpful to check the osmolality of all the extracellular solutions, even though they were nominally targeted to be iso-osmotic.(2) Some more details about the experimental design and the results are needed from Figure 1:(i) For how long are the cells serum-starved? From the Methods section, it seems like the G1 release in different K+ concentration is done without serum, is this correct? Is the prior thymidine treatment also performed in the absence of serum?(ii) There is a question of whether depolarization constitutes a physiologically relevant mechanism to regulate proliferation, and how depolarization interacts with other extracellular signals that might be present in an in vivo context. Does depolarization only promote proliferation after extended serum starvation (in what is presumably a stressed cell state)? What fraction of total cells are observed to be mitotic (without normalization), and how does this compare to the proliferation of these cells growing in serum-supplemented media? Can K+ concentration tune proliferation rate even in serum-supplemented media?(3) In Figure 2, there are some possible concerns with the perfusion experiment:(i) Is the buffer static in the period before perfusion with high K+, or is it perfused? This is not clear from the Methods. If it is static, how does the ERK activity change when perfused with 5 mM K+? In other words, how much of the response is due to flow/media exchange versus change in K+ concentration?(ii) Why do there appear to be population-average decreases in ERK activity in the period before perfusion with high K+ (especially in contrast to Fig. 3)? The imaging period does not seem frequent enough for photobleaching to be significant.(4) Figure 3 contains important results on couplings between membrane potential and MAPK signaling. However, there are a few concerns:(i) Does cell volume change upon voltage clamping? Previous authors have shown that depolarizing voltage clamp can cause cells to swell, at least in the whole-cell configuration: https://www.cell.com/biophysj/fulltext/S0006-3495(18)30441-7 . Could it be possible that the clamping protocol induces changes in ERK signaling due to changes in cell volume, and not by an independent mechanism?(ii) Does the -80 mV clamp begin at time 0 minutes? If so, one might expect a transient decrease in sensor FRET ratio, depending on the original resting potential of the cells. Typical estimates for resting potential in HEK293 cells range from -40 mV to -15 mV, which would reach the range that induces an ERK response by depolarizing clamp in Fig. 3B. What are the resting potentials of the cells before they are clamped to -80 mV, and why do we not see this downward transient?(5) The activation of ERK by perforated voltage clamp and by high extracellular K+ are each convincing, but it is unclear whether they need to act purely through the same mechanism - while additional extracellular K+ does depolarize the cell, it could also be affecting function of voltage-independent transporters and cell volume regulatory mechanisms on the timescales studied. To more strongly show this, the following should be done with the HEK cells where there is already voltage clamp data:(i) Measure resting potential using the perforated patch in zero-current configuration in the high K+ medium. Ideally this should be done in the time window after high K+ addition where ERK activation is observed (10-20 minutes) to minimize the possibility of drift due to changes in transporter and channel activity due to post-translational regulation.(ii) Measure YFP/CFP ratio of the HEK cells in the high K+ medium (in contrast to the U2OS cells from Fig. 2 where there is no patch data).(iii) The assertion that high K+ is equivalent to changes in Vmem for ERK signaling would be supported if the YFP/CFP change from K+ addition is comparable to that induced by voltage clamp to the same potential. This would be particularly convincing if the experiment could be done with each of the 15 mM, 30 mM, and 145 mM conditions.(6) Line 170: "ERK activity was reduced with a fast time course (within 1 minute) after repolarization to -80 mV." I don't see this in the data: in Fig. 3C, it looks like ERK remains elevated for > 10 min after the electrical stimulus has returned to -80 mVComments on revisions:The authors have done a good job addressing the comments on the previous submission.
**Reviewer #3 (Public review):**
Summary:This paper demonstrates that membrane depolarization induces a small increase in cell entry into mitosis. Based on previous work from another lab, the authors propose that ERK activation might be involved. They show convincingly using a combination of assays that ERK is activated by membrane depolarization. They show this is Ca2+ independent and is a result of activation of the whole K-Ras/ERK cascade which results from changed dynamics of phosphatidylserine in the plasma membrane that activates K-Ras. Although the activation of the Ras/ERK pathway by membrane depolarization is not new, linking it to an increase in cell proliferation is novel.StrengthsA major strength of the study is the use of different techniques - live imaging with ERK reporters, as well as Western blotting to demonstrate ERK activation as well as different methods for inducing membrane depolarization. They also use a number of different cell lines. Via Western blotting the authors are also able to show that the whole MAPK cascade is activated.WeaknessesA weakness of the study is the data in Figure 1 showing that membrane depolarization results in an increase of cells entering mitosis. There are very few cells entering mitosis in their sample in any condition. This should be done with many more cells to increase the confidence in the results. The study also lacks a mechanistic link between ERK activation by membrane depolarization and increased cell proliferation.The authors did achieve their aims with the caveat that the cell proliferation results could be strengthened. The results, for the most par,t support the conclusions.This work suggests that alterations in membrane potential may have more physiological functions than action potential in the neural system as it has an effect on intracellular signalling and potentially cell proliferation.In the revised manuscript, the authors have now addressed the issues with Figure 1, and the data presented are much clearer. They did also attempt to pinpoint when in the cell cycle ERK is having its activity, but unfortunately, this was not conclusive.
**Reviewer #2 (Recommendations for the authors):**
Small issues:Fig. 1A. Please add a mark on the timeline showing when the K+ concentration is changed. Also, please add a time axis that matches the time axis in (C), so readers can know when in C the medium was changed.1B caption: unclear what "the images were 20 min before and after cytokinesis" means, given that the images go from -30 min to +20 min. Maybe the authors mean, "the indicated times are measured relative to cytokinesis."

Thank you for bringing these points to our attention that can confuse readers. We revised the figure legend.

Line 214: nonoclusters  nanoclustersLine 475: 10 mm -> 10 ¥mum

Corrected.